# Behavioural Differences in Sensorimotor Profiles: A Comparison of Preschool-Aged Children with Sensory Processing Disorder and Autism Spectrum Disorders

**DOI:** 10.3390/children9030408

**Published:** 2022-03-14

**Authors:** Giulia Purpura, Francesco Cerroni, Marco Carotenuto, Renata Nacinovich, Luca Tagliabue

**Affiliations:** 1School of Medicine and Surgery, University of Milano Bicocca, 20900 Monza, Italy; renata.nacinovich@unimib.it; 2Clinic of Child and Adolescent Neuropsychiatry, Università degli Studi della Campania “Luigi Vanvitelli”, 81100 Caserta, Italy; francesco.cerroni@unicampania.it (F.C.); marco.carotenuto@unicampania.it (M.C.); 3Child and Adolescent Health Department, San Gerardo Hospital, ASST of Monza, 20900 Monza, Italy; luca.tagliabue@unimib.it

**Keywords:** sensory processing disorder, autism spectrum disorder, neurodevelopmental disorders, repetitive movements, sensory processing measures, pre-schoolers

## Abstract

Sensory Processing Disorders (SPDs) define dysfunctions in modulating, organising, and using information from several sensory channels for regulating motor, behavioural, emotional and attention responses. Although SPD can be identified also as an isolated condition in young children, its presence in Autism Spectrum Disorder (ASD) population is really frequent. The study purpose is to explore the SPD clinical expression and the putative correlation with several behavioural aspects both in children with ASD and in those with isolated SPD. Therefore, 43 preschool-aged children (25 ASD vs. 18 SPD) were recruited, and their parents completed three questionnaires (Developmental Profile-3, Sensory Processing Measure–Preschool, Repetitive Behaviour Scale-Revised) to evaluate behavioural alterations and developmental levels. The main result is that both ASD and SPD groups had significantly sensory-related behavioural symptoms, although ASD children seem to be more impaired in all areas. Several significant correlations were found between sensory processing difficulties and repetitive behaviours, but in the SPD group a specific relationship between Body Awareness and Ritualistic/Sameness Behaviour was found. Conversely, in the ASD group, more diffuse interlinks between sensory processing difficulties and motor behaviours were significant. In conclusion, the present study confirms the key role of sensory–motor skills in early diagnosis and intervention among children at risk for neurodevelopmental disorders.

## 1. Introduction

Sensory Processing Disorders (SPD) are a group of dysfunctions in modulating, organising and using information from several sensory channels to regulate motor, behavioural, emotional and attentional responses for environment adaptation [1,2]. Children with SPD can show several and early behavioural problems, such as irritability, abdominal colic, sleep–wake cycles disorders, and difficulties in social communication development. Moreover, children with SPD tend to cry easily and to manifest exaggerated emotive reactions [3]. According to the Diagnostic Classification of Mental Health and Developmental Disorders of Infancy and Early Childhood (DC:0-5) [4], SPDs can be diagnosed when the infant/young child shows abnormalities in regulating input in one or more sensory domains (e.g., tactile, visual, auditory, vestibular, olfactory, taste, proprioceptive and interoceptive), in the absence of other major physical, neurological or psychiatric conditions.

SPD symptoms can be heterogeneous and characterised by sensory stimulus over-responsivity (i.e., persistent pattern of exaggeration, intense or prolonged response), under-responsivity (i.e., reduced and slow responsiveness), or atypical responses (i.e., extended exploration of stimuli because it is not properly recognised) [4]. Consequently, SPD can impact self-regulation, influencing behaviour, daily-life activities, learning and neuropsychomotor development [1,5,6]. Recently, Mulligan [7] suggested that adaptive behaviour can be frequently altered by sensory processing impairment during childhood and that the severity of SPD seems to be negatively correlated with adaptive skills.

Although SPDs can be present also as isolated clinical conditions from early infancy to preschool age, their association with specific neurodevelopmental disorders is frequent. Particularly, the sensory processing dysfunction associated with specific social communicative deficits supports the Autism Spectrum Disorder (ASD) diagnosis [8]. As a matter of fact, according to DSM-5 and DC:0-5^TM^, ASD is characterised by impaired social communication, restricted interests and repetitive behaviours, including the atypical sensory issues [4,8].

Moreover, among children with ASD, sensory processing problems seem to be closely associated with motor impairment and repetitive movements starting from the first years of life [9,10], while among children with a high risk of ASD, the early reduced variability and flexibility of sensorimotor systems and the atypical transition between behavioural states from birth may justify the later difficulties to obtain important cues from the environment’s sensory inputs, and this dysfunction may precede and contribute to an atypical brain development trajectory in toddlerhood [11]. In 2012, Pellicano and Burr [12] proposed the hypothesis that the atypicalities in the sensory processing of ASD subjects could be an effect of hypo-priors, or reduced experiential bias. According to this idea, the lack of modulated experiences by prior knowledge from the environment may flow in a reduced generalisation, which in turn could constrain motor plans to those previously acquired. In this light, in the absence of the moderating effect of priors, repetitive behaviours may be interpreted as way to reduce the environmental uncertainty.

In this perspective, the main goal of the present study is exploring the presence of SPD and the putative correlation with behavioural aspects both among children with ASD and in non-ASD children with SPD. Our hypothesis is finding differences for the severity of sensory processing difficulties between the two groups with different effects on developmental trajectories.

## 2. Materials and Methods

### 2.1. Sampling and Data Collection

The present observational study was performed in the Child Neuropsychiatry Unit of San Gerardo Hospital of Monza (Italy) at the University of Milano Bicocca (Italy), and in the Clinic of Child and Adolescent Neuropsychiatry at the University of Campania “Luigi Vanvitelli” in Naples (Italy). Ethical approval was not needed, considering that SPD evaluation is routinely performed in the clinical practice for all the clinical units participating. Moreover, the data were collected anonymously and signed informed consent from parents was obtained. The protocol study was carried out in accordance with the Declaration of Helsinki.

Therefore, 43 children (18 with isolated SPD and 25 with ASD) were recruited and their parents filled out three validated tools questionnaires (see Section 2.2) to assess the psychomotor profiles. The inclusion criteria for the study were as follows: (i) children with SPD or ASD were assessed by a multidisciplinary team, according to DSM-5 and DC:0-5 criteria; (ii) the age of the children had to be between 2 and 5 years; (iii) the absence of major sensory impairments in the child. The exclusion criteria were as follows: (i) children with genetic, neurological, or psychiatric conditions; (ii) children with epilepsy or seizures controlled by pharmacotherapy; (iii) children with parents who do not speak the Italian language.

Specifically, 25 subjects with ASD (males: 19; females: 6) and 18 subjects with isolated SPD (males: 16; females: 2) were recruited. The main characteristics of the sample are shown in Table 1.

### 2.2. Measures

#### 2.2.1. Developmental Profile 3 (DP-3)

DP-3 [13] assesses developmental function in five developmental domains for children aged from birth to 12 years old. These five domains include cognitive functioning, communication, social–emotional development, adaptive behaviour and motor development, assessed using a standardised 180-item checklist in two forms: (1) Questionnaire Form, independently completed by parents, and (2) Interview Form, completed by parents with the help of clinicians. Additionally, the combination of the scores obtained from these five developmental domains also was used to calculate a general development standard score (GDS), which represents the overall development level of a child. For this study, the Questionnaire Form in the Italian Version, with Italian normative data, was used [14].

#### 2.2.2. Sensory Processing Measure-Preschool (SPM-P)

The SPM-P is a rating scale with two forms—the Home Form and the School Form—each with 75 items for the parent/caregiver and teacher/daycare provider, respectively, to complete. The items on the forms are designed for children from 2 to 5 years old and cover a wide range of behaviours and characteristics related to sensory processing and functional performance [15]. In our study, only the Home Form was administered in the published Italian version and using Italian normative data [16]. The SPM-P provides eight scaled scores: Vision, Hearing, Touch, Body Awareness (non-technical term for proprioception), Balance and Motion (non-technical term for vestibular), Total Sensory System Score, Planning and Ideas and Social Participation. A lower score indicates better sensory processing function. A score between 60 and 69 (some differences) indicates a borderline range of problems in sensory processing, while a score of 70 or above (definite difference) indicates a significant sensory processing and integration problem that may noticeably affect the child’s daily functioning (equates to a score +2.0 SD above the norm).

#### 2.2.3. Repetitive Behaviour Scale-Revised (RBS-R)

The RBS-R is a questionnaire proposed by Bodfish and collaborators [17] that provides a detailed assessment of repetitive movements and interests in children. In our study the Italian Version of the RBS-R was used [18]. The RBS-R is an empirically derived clinical rating scale fulfilled by parents, which provides a quantitative, continuous measure of the full spectrum of repetitive behaviours. Items are rated on a four-point Likert scale (ranging from 0 = behaviour does not occur to 3 = behaviour occurs often and is a severe problem). The scoreable items on the RBS-R are grouped into five subscales: (1) Stereotyped Behaviour; (2) Self-Injurious Behaviour; (3) Compulsive Behaviour; (4) Ritualistic Behaviour/Sameness Behaviour; and (5) Restricted Interests Behaviours. Two raw scores can be calculated for each subscale; one based on the summed item scores within each subscale named ‘‘Score’’, and one based on the number of items endorsed (i.e., number of items with non-zero score) named ‘‘Endorsement’’. The “Total Score” (the sum of all subscale scores) and the “Total Number-Endorsed” were also calculated.

### 2.3. Statistical Analysis

Preliminary standard statistical tests (the non-parametric Mann–Whitney U test for two independent populations) were employed to check the comparability of the two groups for gender, age and gestational age at birth. A comparison of the SPD group to the ASD group for clinical continuous factors (SPM-P, RBS-R and DP-3) was performed by using an independent T-test. Descriptive analyses were reported where appropriate.

A two-tail bivariate parametric correlation test (Pearson Test) between age and gestational age with DP-3 (GDS and subscales), SPM-P (Total and subscales) and RBS-R (Total and subscales) was carried out. Two-tailed partial correlation analyses, controlling for age and for GDS of DP-3, were performed to verify the relationship between sensory processing (SPM-P) and repetitive movements (RBS-R), both in the whole sample and separately in the two groups.

Finally, two-tailed partial correlation analyses, controlling for age and groups (SPD and ASD), were performed to evaluate the possible links between functioning profile (DP-3), and sensorial and behavioural domains (SPM-P and RBS-R) in the whole sample.

A *p*-value below 0.05 was interpreted as significant. All analyses were achieved using SPSS 20.0 software.

## 3. Results

### 3.1. Differences between the Two Groups

Preliminary standard statistical tests showed the absence of significant differences between the two groups, confirming the homogeneity of the sample in terms of gender, age and perinatal condition. The independent T-test between groups showed a significant difference in the SPM-P-Total Score (*p* < 0.001), in the RBS-R-Total Score (*p* < 0.001), in the RBS-R-Total Number-Endorsed, and in the DP-3-GDS (*p* = 0.003), showing a bigger deficit in all investigated domains in the ASD group (see Table 2).

As regards sensory processing abilities, it is evident that the mean scores of the SPD group are near or into the borderline range (t-point scores from 60 to 69) for the Italian normative data, while for the ASD group, the scores are for the most part in the range of disorder (t-point scores from 70). For both groups, the most impaired sensorial domain seems to regard the Planning and Ideas subscale (SPD mean score: 62.7; ASD mean score: 75.2) (see Figure 1).

### 3.2. Correlations in the Whole Sample

No significant correlations were found between age and sensory processing scores (SPM-P), repetitive movements (RBS-R) and developmental profile (DP-3), considering Total Scores and Subscales Scores. Similarly, no significant correlations with the gestational age at birth were found.

Considering the whole sample, the two-tailed partial correlation analysis weighted for age of children and GDS at DP-3, showed a significant correlation between the Total Sensory System Score of SPM-P and Stereotyped Behaviour Score (rho = 0.400, *p* = 0.010), Stereotyped Behaviour Endorsement (rho = 0.434, *p* = 0.005), Ritualistic/Sameness Behaviour Score (rho = 0.347, *p* = 0.026), Ritualistic/Sameness Behaviour Endorsement (rho = 0.384, *p* = 0.016), Restricted Interests Score (rho = 0.354, *p* = 0.023), Restricted Interests Endorsement (rho = 0.345, *p* = 0.027), RBS-R Total Score (rho = 0.413, *p* = 0.007) and RBS-R Total Number Endorsed (rho = 0.421, *p* = 0.006). Moreover, we found a significant correlation between SPM-P-Vision and Stereotyped Behaviour Score (rho = 0.404, *p* = 0.009), Stereotyped Behaviour Endorsement (rho = 0.407, *p* = 0.008), and RBS-R Total Score (rho = 0.349, *p* = 0.025). As regards SPM-P-Hearing, significant correlations with Stereotyped Behaviour Score (rho = 0.377, *p* = 0.015), Stereotyped Behaviour Endorsement (rho = 0.374, *p* = 0.016), Ritualistic/Sameness Behaviour Score (rho = 0.360, *p* = 0.021), Restricted Interests Score (rho = 0.454, *p* = 0.003), Restricted Interests Endorsement (rho = 0.335, *p* = 0.032), RBS-R Total Score (rho = 0.432, *p* = 0.005) and RBS-R Total Number Endorsed (rho = 0.342, *p* = 0.029) were found. Significant correlations were also found between SPM-P-Touch and RBS-R Stereotyped Behaviour Score (rho = 0.437, *p* = 0.004), RBS-R Stereotyped Behaviour Endorsement (rho = 0.473, *p* = 0.002), RBS-R Self-Injurious Behaviour Score (rho = 0.422, *p* = 0.006), RBS-R Self-Injurious Behaviour Endorsement (rho = 0.379, *p* = 0.015), Ritualistic/Sameness Behaviour Score (rho = 0.384, *p* = 0.013), Ritualistic/Sameness Behaviour Endorsement (rho = 0.403, *p* = 0.009), Restricted Interests Score (rho = 0.358, *p* = 0.022), RBS-R Total Score (rho = 0.476, *p* = 0.002) and RBS-R Total Number Endorsed (rho = 0.465, *p* = 0.002). Moreover, there were significant correlations between SPM-P-Planning and Ideas and Stereotyped Behaviour Endorsement (rho = 0.325, *p* = 0.038), Ritualistic/Sameness Behaviour Score (rho = 0.321, *p* = 0.040), Restricted Interests Score (rho = 0.352, *p* < 0.001), Restricted Interests Endorsement (rho = 0.415, *p* = 0.007), RBS-R Total Score (rho = 0.384, *p* = 0.013) and RBS-R Total Number Endorsed (rho = 0.327, *p* = 0.037).

Finally, in the whole sample, we found significant correlations between developmental profile (DP-3 Total and Subscales) and sensory-related behavioural symptoms (SPM-P) and repetitive movements (RBS-R), as reported in Table 3.

### 3.3. Correlation within the Two Groups

Considering only the SPD group (see Figure 2), some significant correlations were found between SPM-P- Body Awareness and Ritualistic/Sameness Behaviour Score (rho = 0.558, *p* = 0.025), Ritualistic/Sameness Behaviour Endorsement (rho = 0.490, *p* = 0.054) and RBS-R Total Score (rho = 0.504, *p* = 0.046).

Regarding the ASD group (see Figure 3), significant correlations confirmed the relationship between Total Sensory System Score of SPM-P and Stereotyped Behaviour Score (rho = 0.467, *p* = 0.025), Stereotyped Behaviour Endorsement (rho = 0.512, *p* = 0.013) and Restricted Interests Endorsement (rho = 0.462, *p* = 0.026); between SPM-P-Vision and Stereotyped Behaviour Score (rho = 0.436, *p* = 0.038) and Endorsement (rho = 0.424, *p* = 0.044); between SPM-P-Hearing and Restricted Interests Score (rho = 0.479, *p* = 0.021); between SPM-P-Touch and Stereotyped Behaviour Score (rho = 0.475, *p* = 0.022), Endorsement (rho = 0.521, *p* = 0.011) and RBS-R Total Score (rho = 0.405, *p* = 0.056); and between SPM-P- Planning and Ideas and Restricted Interests Score (rho = 0.694, *p* < 0.001) and Restricted Interests Endorsement (rho = 0.543, *p* = 0.007).

## 4. Discussion

This study compared the behavioural reactions to the sensory stimulation of children with ASD and with isolated SPD, utilising and integrating scores of three different types of parent-report questionnaires (DP-3, SPM-P, RBS-R). The main result of the present report can be summarised as follows: both ASD and SPD groups tend to present significantly sensory-related behavioural symptoms, although psychomotor similarities and differences were observed between the two groups regarding the influences of these in daily-life activities.

Specifically, as expected, the ASD group showed a severe alteration in sensory-related behaviours (SPM-P), in the repetitive movements pattern (RBS-R), and in the functioning profile (DP-3). Moreover, in both groups the most impaired sensorial domain was the Planning and Ideas Subscale, referring to the integrative functions for translating sensory inputs into planning, programming and performing finalised motor actions. In the whole sample, this subscale of SPM-P seems to be linked to the Social–Emotional Subscale of the DP-3 scale, but also with several repetitive behaviours at RBS-R, especially with the Restricted Interests. Again, the Motor Development Subscale of DP-3 appears to be related to different sensory and behavioural symptoms, which include Hearing, Touch, Balance and Motion and Social Participation in the SPM-P questionnaire, but also Stereotyped Behaviour in the RBS-R questionnaire, suggesting that poorer performance was related to a greater atypia in sensory processing and to a higher frequency and intensity of repetitive and stereotyped behaviours. Similarly, the correlation between abnormal scores in the Touch Subscale and the frequency and intensity of Self-Injurious Behaviours suggests some possible explanation for the problem behaviours of these children during daily-life activities. Taken as a whole, these data highlight the association between levels of participation and sensorimotor processing skills in childhood that can negatively impact on developmental trajectories from the first years of life. Regarding this, Hertzog et al. [19] suggested that typically developing children participate more than children with SPD in physically active experiences from infancy, which encourages the development of sensory processing and provides a foundation for an ongoing process of acquiring increasing abilities to meet environmental demands.

Another important result of this study is the different relationship between the processing of sensory inputs and the several types of behavioural motor problems, independently from age and developmental levels. As a matter of fact, in young children with SPD, there is a specific relationship between the proprioceptive processing and the Ritualistic/Sameness Behaviours, while in young children with ASD, vision, touch and hearing processing appear correlated to Stereotyped Behaviour and Restricted Interests.

Conversely, we could hypothesise that among SPD children, the difficulties concerning their bodily spatial awareness and postural control, may influence behaviour, leading them to prefer stable and routine situations. Instead, in ASD children the severe complex sensory processing impairment appears to impact the motor variability necessary to integrate the perception with the action and for anticipating and controlling the movement in a well-coordinated way. In this light, processing differences in vision, hearing and touch of children with ASD has been largely described in literature and several authors tend to agree that sensory difficulties are at the basis of poor motor variability in paediatric age [9,20,21,22].

On the other hand, Hadders-Algra [23] suggested the association between the reduction in long-distance cortical connectivity with a limited repertoire of motor behaviour in ASD children. Moreover, this report may be considered as in line with the findings of other studies that highlighted the association between stereotyped behaviours and the degree of motor impairment in school-aged children with ASD [24,25,26]. As a matter of fact, it was widely reported that ASD is a multidimensional condition in which impairments in multisensory and sensory–motor integration reflect several differences in brain connectivity and in developmental age [11,27,28].

In this scenario, ASD would emerge not as a higher-order social–cognitive deficit, but because of an impairment of the primordial ability to process low level sensory, motor and perceptual information gained through experience with other people since the earliest periods of life [29]. This idea may also be sustained by several studies that reported the presence of early sensorimotor signs in infants/young children with, or at risk of, ASD [30,31,32], and also the presence of a relationship between early motor delay and later communication delay in infants at risk for ASD [33,34]. Unlike the comparison group, the ASD children appeared more likely to have abnormalities across multiple sensory domains, whereas most of the comparison group children had weaker abnormal sensory features. Already, Leekam et al. [35] has indicated that sensory abnormalities are more prevalent in children with autism than in children with other developmental disabilities, independently from the cognition level, and, as suggested by Mulligan and collaborators [7], adaptive behaviour across functional domains may be impacted by the severity of sensory processing deficits of the children.

Our results highlighted the presence of altered sensory patterns in the two groups, showing how the abnormalities in the sensory afferents and the lack in integrating them may have cascading effects on all aspects of the neurodevelopment, including detail perception, motor planning and social interaction. Moreover, the correlations between the tests scores confirm the starting hypothesis about the influence of sensory functioning on the behavioural domain, which requires the subject to make a greater adaptive effort and is not always achievable independently, adopting dysfunctional communication behaviour or trying to keep homeostasis through self-stimulating sensorimotor patterns [36].

Based on these findings, it is necessary to reflect on the importance of the support networks for facilitating the subject and his/her family, to adapt and to integrate themself in the social context. According to this idea, the early assessment of sensorimotor competences in children at risk of neurodevelopmental disorders could be the key to programming and implementing individualised intervention. Furthermore, rehabilitative intervention for these children may be necessarily early, multidisciplinary, family-centred and based on a multisensory integration approach, in which professionals specifically focussed on Neurodevelopmental Disorders Therapy can promote the integrated and harmonious development of the child’s function in several areas [28,37,38,39].

## 5. Limitations

There were certain limitations to our study. These included the exclusive use of parent questionnaires and the small sample size. It would have been desirable to perform a neurodevelopmental assessment integrating the use of more objective tools, administered by professionals, to better investigate the differences between the two groups and the impact of sensory processing disorders on social participation. Moreover, the small sample size limited us to studying factors predictive of unfavourable neurodevelopmental outcomes. Nevertheless, the interesting results obtained could be a starting point for better investigating the relationship between early sensory–motor abnormalities in children with and without neurodevelopmental disorders and the possible implication for early intervention, so we believe that the limitations of this study are outweighed by its originality.

## 6. Conclusions

The results of this study reported the analysis of sensory–motor abnormalities and the correlation with developmental functioning, and the frequency and intensity of repetitive behaviours in children with SPD and in others with ASD. This study pinpoints the likelihood of a variety of sensorial and motor features being present with each of these diagnoses, while suggesting that several differences in possible developmental trajectories are evident from the first years of life. Although this study did not examine any organic or genetic factors, already May-Benson and collaborators [2] have suggested that for many individuals, the potential interplay of genetic factors with pre-, peri- and post-natal events may create the conditions necessary to produce SPD and/or ASD, and even Ayres [40,41] has already hypothesised that genetic factors may make the brain more vulnerable than usual related to the processing of sensory stimuli from the environment.

In conclusion, the early assessment of sensory–motor abilities must be considered as crucial in follow-up programs for children at risk of neurodevelopmental disorders. For these reasons, early interventions programs must also take into consideration this developmental domain, using a communicative–relational approach that involves the body and its movement. According to this idea, sensory–motor skills must be improved in order to permit the knowledge and exploration of the environment and the interaction with it. Finally, supporting the child in the acquisition, learning and generalisation of the skills necessary to orient himself and to be an active operative subject in his own life context could be a keystone in changing the developmental trajectories of this population.

## Figures and Tables

**Figure 1 children-09-00408-f001:**
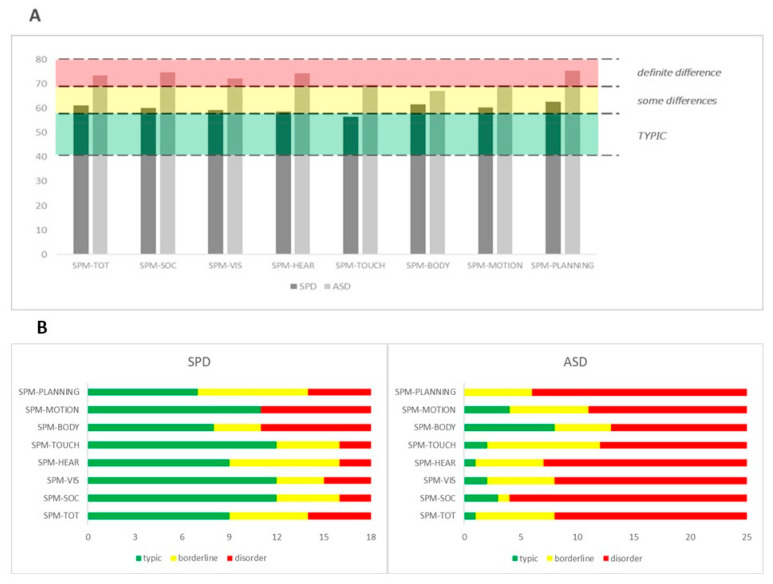
(**A**) SPM-P standard scores in the two groups; (**B**) Percentages of children that have scores in the normal, borderline, or impaired range among the SPM-P subscales in the two groups.

**Figure 2 children-09-00408-f002:**
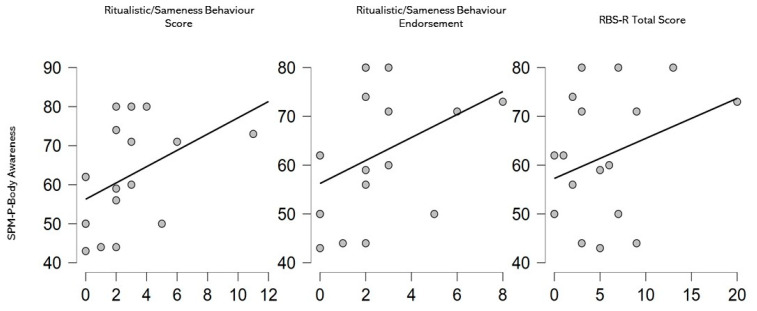
Graphic representations of the significant correlations in the SPD group.

**Figure 3 children-09-00408-f003:**
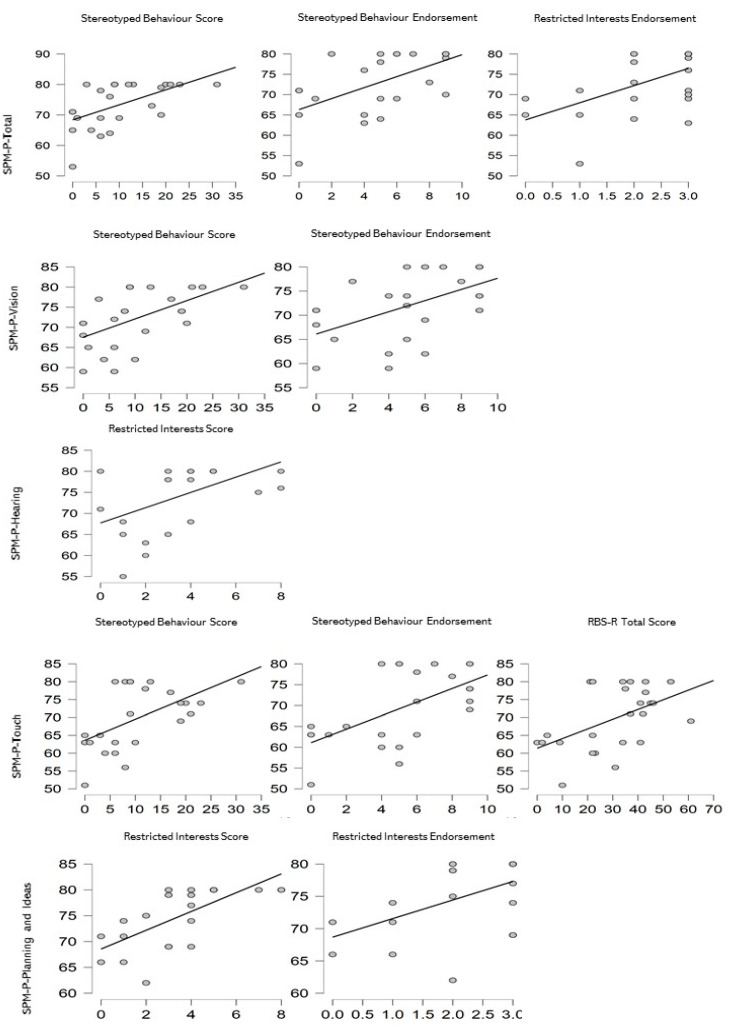
Graphic representations of the significant correlations in the ASD group.

**Table 1 children-09-00408-t001:** Principal characteristics of the sample and results of descriptive statistical analysis (non-parametric Mann–Whitney Test).

	SPD (*n* = 18)	ASD (*n* = 25)	*p*-Value
Gender (*M*; *F*)	16; 2	19; 6	0.290
Age in months (*mean*, *DS*)	53.66 (12.7)	52.7 (9.97)	0.824
Age range (*months*)	31–70	39–71	
Gestational Age in weeks (*means*, *DS*)	38.05 (2.15)	37.68 (2.42)	0.401
Gestational Age range (*weeks*)	30–40	30–40	

**Table 2 children-09-00408-t002:** SPM-P scores, RBS-R scores and DP-3 scores in the two groups.

	SPD	ASD
	*Mean* (*DS*)	*Mean* (*DS*)
SPM-P Total Score	61.2 (9.7)	73.4 (7.4)
SPM-P Vision	59.2 (8.9)	72.2 (7)
SPM-P Hearing	58.7 (9.7)	74.4 (7.6)
SPM-P Touch	56.5 (9.3)	69.6 (8.7)
SPM-P Body Awareness	61.6 (13.03)	67.04 (12.7)
SPM-P Balance and Motion	60.3 (15.8)	69.5 (11.3)
SPM-P Planning and Ideas	62.7 (10.1)	75.2 (5.7)
SPM-P Social Participation	59.7 (8.8)	74.7 (11.2)
RBS-R Stereotyped Behaviour Score	1.1 (1.8)	10.2 (8.4)
RBS-R Stereotyped Behaviour Endorsement	0.83 (1.1)	5.2 (3.2)
RBS-R Self-Injurious Behaviour Score	0.3 (0.75)	1.9 (1.8)
RBS-R Self-Injurious Behaviour Endorsement	0.22 (0.54)	1.6 (1.5)
RBS-R Compulsive Behaviour Score	0.7 (0.9)	4 (3.1)
RBS-R Compulsive Behaviour Endorsement	0.7 (0.9)	2.5 (1.6)
RBS-R Ritualistic/Sameness Behaviour Score	2.5 (2.7)	10.6 (5.6)
RBS-R Ritualistic/Sameness Behaviour Endorsement	2.3 (2.1)	6.9 (3.2)
RBS-R Restricted Interests Score	0.7 (0.8)	3.7 (2.2)
RBS-R Restricted Interests Endorsement	0.7 (0.8)	2.3 (0.9)
RBS-R Total Score	5.3 (5.1)	30.32 (16.3)
RBS-R Total Number Endorsed	4.7 (4.1)	18.5 (8.5)
DP-3 General Developmental Standard Score	60.8 (22.02)	52.7 (9.9)
DP-3 Motor Development	82.3 (21.72)	32.3 (13.9)
DP-3 Adaptive Behaviour	68.5 (17.03)	49.9 (15.6)
DP-3 Social-Emotional Development	69.7 (20.9)	35.4 (17.5)
DP-3 Cognitive Functioning	65.9 (22.2)	37 (18)
DP-3 Communication	61.1 (20.9)	36.8 (17.6)

**Table 3 children-09-00408-t003:** There was a significant correlation between DP-3 scores and SPM-P and RBS-R scores in the whole sample.

DP-3 Scores	Significant Correlation, Controlling for Age and Groups	
DP-3 General Developmental Standard Score	SPM-P	Social Participation (rho = −0.567; *p* < 0.001); Vision (rho = −0.339; *p* = 0.030).
	RBSR	Stereotyped Behaviour Score (rho = −0.329; *p* = 0.036); Stereotyped Behaviour Endorsement (rho = −0.383; *p* = 0.013); Restricted Interests Endorsement (rho = −0.310; *p* = 0.048).
DP-3 Motor Development	SPM-P	Total Score (rho = −0.383; *p* = 0.013); Social Participation (rho = −0.535; *p* < 0.001); Hearing (rho = −0.340; *p* = 0.029); Touch (rho = −0.402; *p* = 0.009); Balance and Motion (rho = −0.402; *p* = 0.047).
	RBS-R	Stereotyped Behaviour Score (rho = −0.309; *p* = 0.050); Stereotyped Behaviour Endorsement (rho = −0.325; *p* = 0.038).
DP-3 Adaptive Behaviour	SPM-P	Social Participation (rho= −0.403; *p* = 0.009).
	RBS-R	Stereotyped Behaviour Score (rho = −0.333; *p* = 0.033); Stereotyped Behaviour Endorsement (rho = −0.319; *p* = 0.042).
DP-3 Social-Emotional Development	SPM-P	Total Score (rho = −0.338; *p* = 0.031); Social Participation (rho = −0.604; *p* < 0.001); Vision (rho = −0.430; *p* = 0.005); Hearing (rho = −0.318; *p* = 0.042); Planning and Ideas (rho = −0.308; *p* = 0.050).
	RBS-R	Stereotyped Behaviour Score (rho = −0.395; *p* = 0.011); Stereotyped Behaviour Endorsement (rho = −0.475; *p* = 0.002); Restricted Interests Score (rho = −0.349; *p* = 0.025); Restricted Interests Endorsement (rho = −0.422; *p* = 0.006); RBS-R Total Score (rho = −0.334; *p* = 0.033); RBS-R Total Endorsed (rho = −0.356; *p* = 0.022).
DP-3 Cognitive Functioning	SPM-P	Social Participation (rho = −0.426; *p* = 0.005); Vision (rho = −0.355; *p* = 0.023); Hearing (rho = −0.343; *p* = 0.028).
	RBS-R	Stereotyped Behaviour Score (rho = −0.339; *p* = 0.030); Stereotyped Behaviour Endorsement (rho = −0.353; *p* = 0.023); Compulsive Behaviour Endorsement (rho = −0.359; *p* = 0.021); RBS-R Total Endorsed (rho = −0.306; *p* = 0.052).
DP-3 Communication	SPM-P	Social Participation (rho = −0.459; *p* = 0.003); Vision (rho = −0.467; *p* = 0.002); Hearing (rho = −0.328; *p* = 0.036).
	RBS-R	Stereotyped Behaviour Score (rho = −0.313; *p* = 0.047); Stereotyped Behaviour Endorsement (rho = −0.306; *p* = 0.052).

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
