# Peer review of "Behavioural Differences in Sensorimotor Profiles: A Comparison of Preschool-Aged Children with Sensory Processing Disorder and Autism Spectrum Disorders"

_children, 2022, doi:10.3390/children9030408_

Round 1

Reviewer 1 Report

In the current manuscript, Giulia et al studied the presence of sensory processing deficits in children with ASD and in children with SPD but not ASD. The authors also explored possible correlations between diverse behavioral aspects in these two cohorts.

The authors reported that both ASD and SPD groups had significant sensory-related behavioral abnormalities, and ASD children particularly were impaired more in all investigated behavioral domains. Significant correlations were found between sensory processing deficits and repetitive behaviors.

The manuscript has been well written, and it was easy to follow in most parts. I don’t have any specific scientific questions. However, I think extensive editing of the manuscript needs to be done as there are several grammatical errors, complex sentences, etc.

Author Response

Thanks for your comments. As suggested, an extensive reviewing of the English was done.

Reviewer 2 Report

I have some questions about enrollment criteria:

  • what are the exclusion criteria?
  • Page 2, lines 87-92: it is confusing, it seems the authors enrolled the parents (and this fact is actually right), but de facto the study investigates children, so the inclusion criteria should regard children.
  • There was no drop out?

Author Response

Thanks for your comments. We clarified the points indicated by you.

  1. We inserted the exclusion criteria in the manuscript (see Materials and Methods Section, from page 2 line 87 to page 3 line 92);
  2. The enrollment methodology was clarified and better explained (see Sampling and data collection Section, pages 2 and 3)
  3. There was not drop out becouse these questionnaires were administered very frequent in our clinical practice.
  4. The English of the manuscript was reviewed.

Round 2

Reviewer 2 Report

Authors well answered to my comments